# Association of Elevated Serum Uric Acid with Nerve Conduction Function and Peripheral Neuropathy Stratified by Gender and Age in Type 2 Diabetes Patients

**DOI:** 10.3390/brainsci12121704

**Published:** 2022-12-12

**Authors:** Wanli Zhang, Lingli Chen, Min Lou

**Affiliations:** 1Department of Neurology, The Second Affiliated Hospital of Zhejiang University, School of Medicine, Hangzhou 310009, China; 2Department of Neurology, The First Affiliated Hospital of Wenzhou Medical University, Wenzhou 325000, China

**Keywords:** diabetic peripheral neuropathy, uric acid, type 2 diabetes, age, gender

## Abstract

Background: The relationship between serum uric acid (SUA) level and diabetic peripheral neuropathy (DPN) remains controversial. We aimed to investigate the association between SUA level and DPN and evaluate the effects of SUA level on nerve conduction function via electromyography in patients with type 2 diabetes (T2DM), stratified by gender and age. Methods: This cross-sectional study included 647 inpatients with T2DM from the First Affiliated Hospital of Wenzhou Medical University between February 2017 and October 2020. The diagnosis of DPN was confirmed according to the Toronto Expert Consensus. Clinical data, SUA level, and nerve conduction parameters were obtained from electronic medical records. Results: A total of 647 patients with T2DM were included, and 471 patients were diagnosed with DPN. The level of SUA was higher in the DPN group than in the Non-DPN group (330.58 ± 99.67 vs. 309.16 ± 87.04, *p* < 0.05). After adjustment, a higher SUA level was associated with the presence of DPN [odds ratio (OR) 1.003, 95% confidence interval (CI), 1.001–1.005; *p* = 0.017]. The area under the curve for the prediction of DPN was 0.558 (95% CI, 0.509–0.608; *p* = 0.022), and the optimized cut-off of SUA level was 297.5 µmol/L. The SUA > 297.5 µmol/L level was independently associated with DPN in the male subgroup (OR 2.507, 95% CI, 1.405–4.473; *p* = 0.002) rather than in the female subgroup. Besides, SUA > 297.5 µmol/L was independently associated with DPN in the younger subgroup (age < 65 years) (OR 2.070, 95% CI, 1.278–3.352; *p* = 0.003) rather than in the older subgroup. In multiple linear regression analysis, SUA was significantly correlated with certain nerve conduction study parameters in the all patients group, and was also observed in the male and younger subgroups. Conclusions: Elevated SUA was independently associated with poorer nerve conduction functions, and hyperuricemia was also significantly associated with a higher risk of developing DPN in T2DM patients, especially in male and younger patients.

## 1. Introduction

Diabetic peripheral neuropathy (DPN), a primary chronic microvascular complication, affects more than half of patients with type 2 diabetes mellitus (T2DM) [1,2]. DPN is frequently associated with falls, pain, functional disability, and diminishing quality of life [3,4]. It is also the most typical cause of foot ulceration and amputation, which leads to an enormous financial burden on families and society [5]. However, the underlying pathophysiological processes of DPN remain not fully understood. Although several factors related to the occurrence of DPN have been recognized [6,7,8,9,10], it is not yet possible to slow the progression of DPN by multifactorial interventions [1]. Therefore, the detection of new risk factors is still necessary.

Serum uric acid (SUA) ≥ 420 µmol/L was defined as hyperuricemia in previous studies [11,12]. The effects of serum uric acid (SUA) levels on DPN remain controversial [13,14,15,16]. A meta-analysis indicated that a higher SUA level was associated with an increased incidence of DPN in diabetic patients [17]. At the same time, another cross-sectional study showed that the risk of large-nerve fiber dysfunction decreased as the SUA levels increased in diabetic patients [18]. Meanwhile, the relationship between SUA and DPN in different age and gender groups is still unknown. Furthermore, previous clinical studies have rarely involved the correlation between SUA and nerve conduction function on electromyography in diabetic patients. Thus, our study aimed to investigate the potential associations of SUA with peripheral nerve conduction and DPN in Chinese patients with T2DM.

## 2. Materials and Methods

### 2.1. Study Population

The cross-sectional study included inpatients with T2DM from the First Affiliated Hospital of Wenzhou Medical University continuously between February 2017 and October 2020. The diagnosis of T2DM was confirmed according to the 2017 criteria of the American Diabetes Association (ADA) [19]. A total of 694 participants with T2DM were included in the study. All participants experienced neurological assessment and nerve conduction studies (NCSs).

According to Toronto Expert Consensus, the definitions of minimal criteria for typical DPN had three levels: possible, probable, and confirmed. The diagnostic criteria for DPN in our study were based on the “confirmed criteria”: a combination of an abnormality of nerve conduction testing and clinical signs or symptoms of neuropathy [20]. The exclusion criteria of DPN were determined as follows: (a) patients with other causes of peripheral neuropathy, (b) patients with severe illness, such as malignant tumor, active infection, severe renal or liver disease, or acute heart failure, (c) patients prescribed any medication that might affect SUA in the last one month, (d) a history of thyroid disease or severe cervical or lumbar radiculopathy. Finally, 647 patients were enrolled in this study, which was approved by the ethics committee of the First Affiliated Hospital of Wenzhou Medical University (No. KY2021-R141) (Figure 1). Written informed consent was obtained from patients or relatives.

### 2.2. Peripheral Neuropathy Assessment

All patients we enrolled underwent a detailed neurological assessment at admission. The neuropathy disability score (NDS) and neuropathy symptom score (NSS) were conducted to confirm the presence of neuropathic deficits and neuropathy symptoms by an experienced neurologist as described in our previous study [21]. The NDS evaluated pinprick sensation, temperature perception, Achilles reflex, and vibration sense [22,23]. The NSS was performed based on a questionnaire used for assessing pain or discomfort in the legs [22,24]. The clinical diagnostic criteria for peripheral neuropathy were an NDS score ≥ 6, or an NDS score of 3–5 with an NSS score ≥ 5 [25].

Two electrophysiological specialists performed nerve conduction testing with an electromyography instrument (Kipoint-4 type, Vidi; NDI-200P + type; Poseidon). The local skin temperature of the participants was maintained between 32 to 33.8 °C during measurement. The distal latency, motor action potential amplitude, and conduction velocity (CV) of the bilateral median, ulnar, tibial, and common peroneal nerves were calculated and recorded. Measurements of sensory nerves were performed for the amplitude and CV of the bilateral ulnar, median and superficial peroneal nerves. In addition, the F-wave latency of the bilateral tibial nerve was determined. The lengthening of F-wave minimum latency and the decrease of amplitude and CV of the peripheral nerve were important and objective parameters of DPN [26,27]. Abnormal nerve conduction studies (NCSs) were considered an abnormality of one or more attributes in two or more nerves [28]. The electrophysiological experts judged the results of the nerve conduction testing. The severe side was used in the study. The mean motor nerve CV(MNCV) was calculated (ulnar nerve motor CV + median nerve motor CV + tibial nerve motor CV + common peroneal nerve motor CV)/4. A similar formula was used to calculate the mean motor nerve amplitude (MNAmp), mean sensory nerve CV (SNCV), and mean sensory nerve amplitude (SNAmp) [16].

### 2.3. Clinical Feature Collection and Laboratory Examination

The clinical data and medical history of T2DM patients were obtained from electronic medical records. We collected baseline data on age, sex, duration of diabetes, smoking history, body mass index (BMI), hypertension, and hyperlipidemia. Blood samples were routinely obtained in the morning hours (range: 06:00–10:00) after overnight fasting (at least 8 h). Laboratory tests including for glycated hemoglobin (HbA1c), serum uric acid (SUA), triglyceride, total cholesterol (TC), high-density lipoprotein cholesterol (HDL-C), low-density lipoprotein cholesterol (LDL-C), thyroid stimulating hormone (TSH), free triiodothyronine (FT3), and free thyroxine (FT4) were conducted. SUA and lipid profiles were measured on a Hitachi 7600 biochemistry autoanalyzer using a standardized enzymatic method.

### 2.4. Statistical Analysis

We used SPSS Statistics, Version 24.0 (SPSS Inc., Chicago, IL, USA), and packages “forestplot” version 2.0.1 and “rms” version 6.2.0 as implemented in R version 4.1.2 (R Foundation for Statistical Computing, Vienna, Austria) for plots and statistical analysis. Two-sided *p* < 0.05 was considered statistically significant. Shapiro–Wilk test was performed to confirm the normality of continuous variables. Continuous variables were described as the medians with interquartile ranges for skewed data or means with standard deviations for normal data. Categorical variables were presented as frequencies and percentages. Statistical analyses were tested by the Kruskal–Wallis test or the Student’s *t*-tests for continuous variables and the chi-square test for categorical variables. Receiver operating characteristic (ROC) curve analyses were performed to determine the predictive ability of SUA for DPN. The SUA level was transformed into a categorical variable based on the optimal cutoff value. Then we divided all patients into two groups by the optimal cutoff value and compared their differences in baseline characteristics. Different multivariate logistic regression models (model 1 and model 2) were used to identify the associations between SUA and DPN. Subgroup logistic regression analyses (model 2) were used, and we tested the statistical significance of covariate category × SUA in the subgroup logistic regression models to explore the multiplicative interaction. We used restricted cubic splines (model 2) with 3 knots at the 10th, 50th, and 90th percentiles to flexibly model and describe relationships between the SUA and the DPN risk in different subgroups. Pearson’s or Spearman’s test and multiple linear regression analysis were, respectively, used to assess the association between SUA and NCSs parameters.

## 3. Results

### 3.1. Baseline Characteristics

The baseline clinical characteristics of 647 patients with type 2 diabetes are given in Table 1. The average age of the total patients was 57.14 ± 13.09 years old, the mean level of SUA was 324.76 ± 96.81 µmol/L, and 62.44% of patients were male. DPN was diagnosed in 471 (72.8%) patients.

### 3.2. Association between SUA Level and the Presence of DPN

Patients were divided into DPN (*n* = 471) and Non-DPN groups (*n* = 176) (Table 1). The DPN group was older and presented a longer duration of diabetes, higher levels of SUA, and higher proportions of smoking and hypertension when compared with the Non-DPN group (*p* < 0.05). Furthermore, compared with the Non-DPN group, the DPN group had a lower level of TC and LDL-C (*p* < 0.05).

The area under the curve (AUC) of SUA for DPN was 0.558 [95% confidence interval (CI), 0.509–0.608; *p* = 0.022], and the optimized cut-off of SUA value was 297.5 µmol/L. The patients were grouped based on the optimized cut-off of SUA value. The clinical and laboratory characteristics of each group are described in Table 2. There were significant differences in sex, BMI, triglyceride, HDL-C, and FT4(*p* < 0.05). The proportion of DPN increased as the SUA value increased.

As presented in Table 3, after adjusting for Model 1 plus variables with *p* value < 0.05 in Table 1 and Table 2 (Model 2), such as BMI, smoking, hypertension, TC, triglyceride, HDL-C, LDL-C, and ft4, patients with SUA level > 297.5 µmol/L showed an increased risk of DPN compared with those with SUA level ≤ 297.5 µmol/L (OR = 1.892; 95% CI 1.225–2.922; *p* < 0.05). Taking the SUA level as a continuous variable, we found that an elevated SUA level was also independently associated with an increased risk of DPN. 

### 3.3. Subgroup Analysis for the Association between SUA Level and the Presence of DPN

In subgroup analysis, our study showed that age and gender might alter the correlation between SUA and DPN (Figure 2). The interaction between serum uric acid and each subgroup variable was not statistically significant. Multivariate logistic regression analyses were performed respectively according to gender and age subgroup (Table 3). In the male subgroup, SUA was an independent risk factor of DPN after adjustment for model 1 and model 2, no matter whether as a continuous variable or a categorical variable., However, this relationship was not found in the female subgroup. In the younger subgroup (age < 65 years), the association between SUA and DPN was significant but it was not significant in the older subgroup (age ≥ 65 years).

To further investigate the correlation between SUA level and DPN, the restricted cubic spline regression with model 2 was visualized in total patients and for different subgroups (Figure 3, Appendix A). The SUA level was linearly associated with DPN in patients with T2DM, and the odds ratio of DPN significantly increased with elevated SUA in the male and younger subgroups.

### 3.4. Relationship between SUA Level and NCSs Parameters

The motor CV of the ulnar nerve and tibial nerve, as well as the MNCV, were slower and the sensory amplitude of the ulnar nerve was lower in the high SUA level group (>297.5 µmol/L) than the group with low SUA level (≤297.5 µmol/L). In contrast, the F-wave minimum latency was significantly longer (*p* < 0.05; Table 4; Appendix A).

Table 5 shows that SUA level was negatively correlated with the motor CV of the ulnar nerve, tibia nerve, common peroneal nerve, the sensory amplitude of the ulnar nerve, and the MNCV. In contrast, it was positively correlated with F-wave minimum latency. Furthermore, there was a significant correlation between serum uric acid and NCSs parameters in the male and younger subgroups.

In multiple linear regression analysis (Table 6), after adjusting for age, sex, duration of diabetes, HbA1c, BMI, smoking, hypertension, TC, triglyceride, HDL-C, LDL-C, and ft4, SUA was significantly correlated with certain NCSs parameters, which was also seen in the male subgroup and the younger subgroup. Appendix A shows the relationship between SUA and NCSs parameters in the female and the older subgroups.

## 4. Discussion

Identifying the specific risk factors associated with the occurrence and progression of DPN is essential. The present cross-sectional study indicates that the elevated SUA levels in patients with T2DM are independently associated with DPN and poor nerve conduction function. Importantly, when the analysis was further stratified by gender and age, the significant associations only existed in male patients and younger patients (age < 65 years) but not in female patients and older patients (age ≥ 65 years).

The relationship between SUA and DPN in T2DM patients remains contradictory [16,17,18,29]. Previous two single-center studies with a small sample observed that the prevalence of DPN increased with SUA level in diabetic patients [16,30]. A meta-analysis, which finally included twelve studies, demonstrated that elevated SUA levels were significantly associated with DPN [17]. On the contrary, Jiang et al. reported an association between lower SUA levels and elevated risk of large-nerve fiber dysfunction in T2DM patients [18]. Another study examining the relationship between SUA and diabetic complications failed to confirm the association between SUA and DPN [29]. The potential reasons for this discrepancy may be due to the different subject designs among these studies. Based on a larger population and the adjustment of more common factors than previous studies, our present study confirms that hyperuricemia was independently associated with the diagnosis of DPN in T2DM patients. 

We also provide a potential answer for the previous controversial results from the subgroup analysis, which may be one of the highlights of our study, as it found that an elevated SUA level was only associated with DPN in the younger or male patients. Similarly, another Chinese study indicated that SUA level was inversely associated with large nerve dysfunctions in the younger male subgroup but not in female or older patients [18]. In other studies, older patients with T2DM have a higher risk of developing DPN [25,31]. The explanation for the SUA level’s failure to promote DPN in older patients was that those older individuals had a longer diabetes duration and were more likely to have comorbidities [31]. The effect of gender on SUA level is that estrogens can promote renal clearance of uric acid [32,33], which may result in the significant association between SUA and DPN in males, but not in females. 

Another strength of our study was confirming the relationship of SUA with nerve conduction function in T2DM patients, as nerve conduction studies (NCSs) are advocated as a gold standard as an early and objective indicator of DPN [20,34]. We observed that slower nerve conduction velocity and longer F-wave minimum latency were mainly associated with increasing SUA levels. It is well known that the pathological characteristics of DPN are progressively demyelination and axonal degeneration [19,20,35]. The phenomenon that the elevated SUA level was associated with reduced nerve conduction velocity suggests that SUA may worsen segmental demyelination. This is consistent with the study results of Lin et al. [16] but opposite to another neuroelectrophysiological study [36]. However, we also further demonstrated that elevated SUA level was independently associated with poor nerve conduction function in the younger or male subgroup but not in female and older patients.

The pathophysiological mechanisms that cause diabetic peripheral neuropathy focus on two of the widest mechanisms: the oxidative stress pathway and the neuroinflammatory pathway [37,38]. Hyperuricemia has been linked to oxidative stress and endothelial dysfunction [39,40], which may predispose to diabetic neuropathy. Elevated serum uric acid decreases the production of nitric oxide, damages endothelial cells, and ultimately leads to microcirculation disturbance [41]. Intracellular uric acid induces pro-inflammatory biomarkers and activates NADPH oxidases, then further increases ROS production which can lead to pro-oxidative actions [42]. Oxidative stress induced by hyperuricemia causes various pathophysiology reactions such as inflammatory cytokine release and cell apoptosis [43]. Uric acid crystals could also activate the inflammasome, leading to the maturation of interleukin-1 and initiation of inflammatory cascades [44]. Thus, these possible mechanisms suggest that uric acid plays a crucial role in developing DPN. Nevertheless, extracellular uric acid is an essential antioxidant against oxidation and scavenging oxygen radicals [45]. This may be why other studies have come to conclusions contrary to ours. A higher or a lower SUA level can induce a range of diseases.

The present study has several remarkable strengths. First, the number of patients included in our study is larger than others. Moreover, this study is the first to determine the possibility of DPN increasing with elevated SUA levels only in the younger or male subgroup. Few studies have evaluated DPN using nerve conduction studies. We also further investigate the relationship between SUA and comprehensive nerve conduction function by gender and age stratification and by multivariate regression analysis. The present study suggests that lowering serum uric acid levels may be beneficial for DPN, especially for male and younger patients.

Nevertheless, the current study has some limitations. Firstly, this was only a single-center study. Thus, the findings may not be generalizable to other centers. Secondly, although collecting data on the most common confounding factors, the study did not adjust for other complications of T2DM. We also did not collect information on hypoglycemic agents for diabetes. Metformin may influence the level of SUA and increase the risk of neuropathy [46]. Thirdly, the nerve conduction testing in our study could only reflect the functional status of large fibrous nerves of peripheral nerves but could not detect the small fibrous nerves. Furthermore, the present study was only focused on type 2 diabetes. The role of SUA in the development of DPN in patients with type 1 diabetes still needs to be paid more attention in the future.

## 5. Conclusions

In conclusion, our study demonstrated that elevated serum uric acid was negatively associated with nerve conduction function, and hyperuricemia was significantly associated with the prevalence of DPN in T2DM patients, especially in male and younger patients. Although further studies are needed to clarify the effect of SUA intervention on DPN development, the study lends support to the viewpoint that special attention should be paid to the male and younger patients with elevated SUA level for their susceptibility to DPN.

## Figures and Tables

**Figure 1 brainsci-12-01704-f001:**
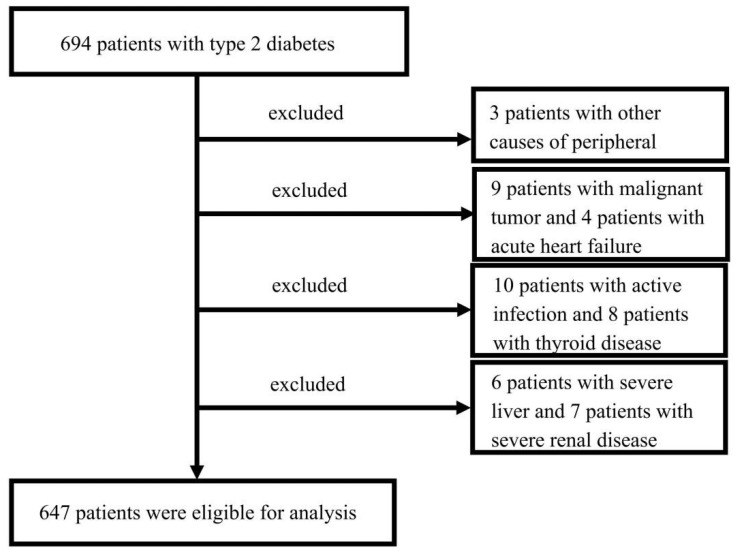
Flow chart showing the patient selection process.

**Figure 2 brainsci-12-01704-f002:**
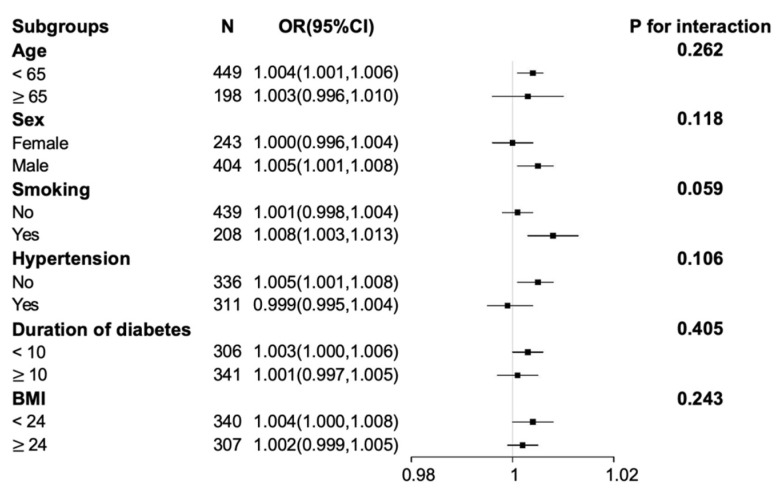
Odds ratios (ORs) for associations between serum uric acid and the risk of diabetic peripheral neuropathy (DPN) in subgroup analyses. The above model (model 2) adjusted for gender, age, duration of diabetes, glycated hemoglobin (HbA1c), body mass index, smoking, hypertension, total cholesterol, triglyceride, high-density lipoprotein cholesterol, low-density lipoprotein cholesterol, and free thyroxine (FT4). In each subgroup, the model is not adjusted for the stratification variable. Abbreviations: CI, confidence interval.

**Figure 3 brainsci-12-01704-f003:**
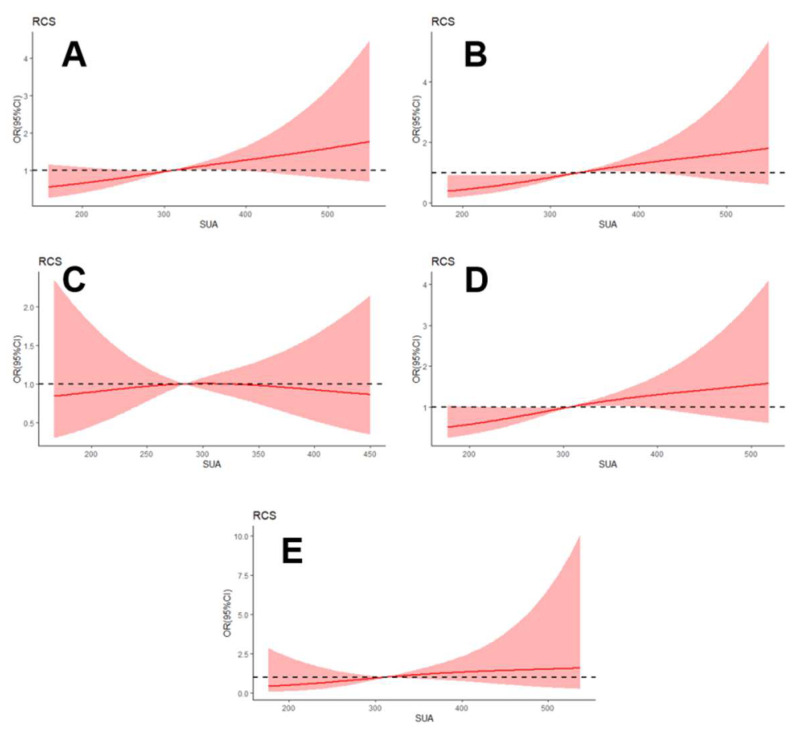
Relationship between serum uric acid level and diabetic peripheral neuropathy in total patients (**A**), male subgroup (**B**), female subgroup (**C**), younger subgroup (**D**), and older subgroup (**E**), respectively, using restricted cubic splines with 3 knots (at the 10th, 50th, and 90th percentiles). The model (model 2) was adjusted for gender, age, duration of diabetes, body mass index, glycated hemoglobin (HbA1c), hypertension, smoking, total cholesterol, triglyceride, high-density lipoprotein cholesterol, low-density lipoprotein cholesterol, and free thyroxine (FT4). In the gender subgroup, the model is not adjusted for the stratification variable. The solid line indicates the odds ratio (OR), while the shadow indicates 95% CIs. The horizontal dashed line is the reference line (OR = 1). Abbreviations: RCS, restricted cubic splines; SUA, serum uric acid.

**Table 1 brainsci-12-01704-t001:** Characteristics of the 647 participants.

Characteristics	Total (*n* = 647)	Non-DPN (*n* = 176)	DPN (*n* = 471)	*p* Value
Age (years)	57.14 ± 13.09	51.26 ± 13.51	59.34 ± 12.23	0.000
Male, *n*. (%)	404 (62.44%)	103 (58.52%)	301 (63.91%)	0.208
Smoking, *n*. (%)	208 (32.15%)	45 (25.57%)	163 (34.61%)	0.028
Hypertension, *n*. (%)	311 (48.07%)	57 (32.39%)	254 (53.93%)	0.000
Hyperlipidemia, *n*. (%)	213 (32.92%)	63 (35.80%)	150 (31.85%)	0.342
Duration of diabetes (years)	10 (4–14)	5 (1–10)	10 (5–17)	0.000
SUA (µmol/L)	324.76 ± 96.81	309.16 ± 87.04	330.58 ± 99.67	0.012
BMI (kg/m^2^)	24.15 ± 3.38	24.51 ± 3.44	24.01 ± 3.35	0.101
HbA1c (%)	9.46 ± 2.34	9.49 ± 2.46	9.46 ± 2.29	0.887
TC (mmol/L)	4.75 ± 1.25	4.93 ± 1.22	4.68 ± 1.26	0.025
TG (mmol/L)	1.89 ± 1.46	1.94 ± 1.48	1.86 ± 1.46	0.539
HDL-C (mmol/L)	1.02 ± 0.29	0.99 ± 0.29	1.03 ± 0.29	0.161
LDL-C (mmol/L)	2.58 ± 0.92	2.73 ± 0.89	2.52 ± 0.92	0.008
TSH (mIU/L)	1.31 (0.86–1.87)	1.31 (0.91–2.20)	1.31 (0.84–1.81)	0.543
FT4 (pmol/L)	11.21 ± 2.30	11.20 ± 2.11	11.22 ± 2.37	0.939
FT3 (pmol/L)	4.77 ± 1.56	4.93 ± 0.72	4.71 ± 1.78	0.107

Diabetic peripheral neuropathy (DPN); SUA, serum uric acid; BMI, body mass index; HbA1c, glycated hemoglobin; TC, total cholesterol; TG, triglyceride; HDL-C, high-density lipoprotein cholesterol; LDL-C, low-density lipoprotein cholesterol; TSH, thyroid stimulating hormone; FT4, free thyroxine; FT3, free triiodothyronine.

**Table 2 brainsci-12-01704-t002:** Comparison of the baseline characteristics between subgroups based on the optimized cut-off of SUA level.

Characteristics	Serum Uric Acid (SUA) Levels		*p* Value
	≤297.5 µmol/L (*n* = 284)	>297.5 µmol/L *n* = 363	
Age (years)	57.13 ± 12.29	57.15 ± 13.69	0.981
Male, *n*. (%)	149 (52.46%)	255 (70.25%)	0.000
Smoking, *n*. (%)	83 (29.23%)	125 (34.44%)	0.159
Hypertension, *n*. (%)	126 (44.37%)	185 (50.96%)	0.096
Hyperlipidemia, *n*. (%)	94 (33.10%)	119 (32.78%)	0.932
Duration of diabetes (years)	10 (4–14)	10 (4–15)	0.812
BMI (kg/m^2^)	23.55 ± 3.02	24.61 ± 3.57	0.000
HbA1c (%)	9.56 ± 2.26	9.39 ± 2.40	0.349
TC (mmol/L)	4.69 ± 1.21	4.80 ± 1.29	0.283
TG (mmol/L)	1.30 (0.92–1.99)	1.69 (1.18–2.35)	0.000
HDL-C (mmol/L)	1.07 ± 0.32	0.99 ± 0.26	0.001
LDL-C (mmol/L)	2.56 ± 0.92	2.59 ± 0.91	0.624
TSH (mIU/L)	1.18 (0.84–1.81)	1.34 (0.89–1.92)	0.082
FT4 (pmol/L)	11.43 ± 2.63	11.04 ± 1.98	0.039
FT3 (pmol/L)	4.85 ± 2.21	4.72 ± 0.73	0.305
DPN, *n*. (%)	191 (67.25%)	280 (77.13%)	0.005

SUA, serum uric acid; BMI, body mass index; HbA1c, glycated hemoglobin; TC, total cholesterol; TG, triglyceride; HDL-C, high-density lipoprotein cholesterol; LDL-C, low-density lipoprotein cholesterol; TSH, thyroid stimulating hormone; FT4, free thyroxine; FT3, free triiodothyronine; DPN, diabetic peripheral neuropathy.

**Table 3 brainsci-12-01704-t003:** Multivariate logistic regression analysis of the association between SUA and DPN.

		Multivariate Adjusted Model (Model 1)		Multivariate Adjusted Model (Model 2)	
		OR (95% CI)	*p*-Value	OR (95% CI)	*p*-Value
All patients	SUA (>297.5 µmol/L vs. ≤297.5 µmol/L)	1.689 (1.126–2.535)	0.011	1.892 (1.225–2.922)	0.004
	SUA as a continuous variable	1.003 (1.000–1.005)	0.026	1.003 (1.001–1.005)	0.017
	SUA Q1 (<257 µmol/L)	Ref.		Ref.	
	SUA Q2 (257–380 µmol/L)	1.162 (0.724–1.865)	0.535	1.173 (0.712–1.935)	0.531
	SUA Q3 (≥380 µmol/L)	1.675 (0.934–3.003)	0.083	1.796 (0.961–3.356)	0.067
Male subgroup	SUA (>297.5 µmol/L vs. ≤297.5 µmol/L)	1.968 (1.161–3.338)	0.012	2.507 (1.405–4.473)	0.002
	SUA as a continuous variable	1.004 (1.001–1.007)	0.012	1.005 (1.001,1.008)	0.004
	SUA Q1 (<257 µmol/L)	Ref.		Ref.	
	SUA Q2 (257–380 µmol/L)	1.461 (0.758–2.816)	0.257	1.510 (0.761–2.994)	0.238
	SUA Q3 (≥380 µmol/L)	2.028 (0.986–4.171)	0.055	2.510 (1.149–5.482)	0.021
Female subgroup	SUA (>297.5 µmol/L vs. ≤297.5 µmol/L)	1.411 (0.741–2.685)	0.295	1.263 (0.623–2.559)	0.517
	SUA as a continuous variable	1.001 (0.997–1.005)	0.691	1.000 (0.996–1.004)	0.953
	SUA Q1 (<257 µmol/L)	Ref.		Ref.	
	SUA Q2 (257–380 µmol/L)	0.957 (0.480–1.910)	0.901	0.880 (0.406–1.908)	0.746
	SUA Q3 (≥380 µmol/L)	1.339 (0.453–3.960)	0.598	0.959 (0.300–3.065)	0.944
Younger subgroup	SUA (>297.5 µmol/L vs. ≤297.5 µmol/L)	1.787 (1.142–2.796)	0.011	2.070 (1.278–3.352)	0.003
	SUA as a continuous variable	1.003 (1.000–1.006)	0.029	1.004 (1.001–1.006)	0.013
	SUA Q1 (<257 µmol/L)	Ref.		Ref.	
	SUA Q2 (257–380 µmol/L)	1.131 (0.671–1.907)	0.644	1.284 (0.739–2.231)	0.375
	SUA Q3 (≥380 µmol/L)	1.665 (0.880–3.151)	0.117	1.897 (0.959–3.755)	0.066
Older subgroup	SUA (>297.5 µmol/L vs. ≤297.5 µmol/L)	1.485 (0.509–4.335)	0.470	1.895 (0.514–6.988)	0.337
	SUA as a continuous variable	1.003 (0.998–1.009)	0.27	1.003 (0.996–1.010)	0.336
	SUA Q1 (<257 µmol/L)	Ref.		Ref.	
	SUA Q2 (257–380 µmol/L)	2.171 (0.607–7.761)	0.233	1.561 (0.302–8.067)	0.595
	SUA Q3 (≥380 µmol/L)	2.581 (0.477–13.963)	0.271	2.061 (0.274–15.484)	0.482

Model 1: adjusted for age, sex, duration of diabetes, and glycated hemoglobin (HbA1c). Model 2: adjusted for Model 1 plus the following variables: body mass index, smoking, hypertension, total cholesterol, triglyceride, high-density lipoprotein cholesterol, low-density lipoprotein cholesterol, and free thyroxine (FT4) that were identified as *p* value < 0.05 in Table 1 and Table 2. SUA, serum uric acid; DPN, diabetic peripheral neuropathy; CI, confidence interval; OR, odds ratio.

**Table 4 brainsci-12-01704-t004:** Comparison of the nerve conduction studies parameters between subgroups based on the optimized cut-off of SUA level.

	SUA Level		*p* Value
	≤297.5 µmol/L (*n* = 284)	>297.5 µmol/L (*n* = 363)	
Motor amplitude (mV)			
Ulnar	12.52 ± 2.62	12.15 ± 3.08	0.103
Median	12.39 ± 3.44	12.25 ± 3.15	0.612
Tibial	13.22 ± 5.81	12.91 ± 6.44	0.531
Common peroneal	6.07 ± 3.34	6.38 ± 3.76	0.276
Motor CV (m/s)			
Ulnar	51.41 ± 5.89	50.28 ± 6.50	0.022
Median	52.36 ± 5.56	51.58 ± 5.08	0.062
Tibial	44.33 ± 4.97	43.38 ± 5.56	0.024
Common peroneal	43.28 ± 4.98	42.70 ± 5.24	0.155
Sensory amplitude (uV)			
Ulnar	34.83 ± 19.86	31.76 ± 19.44	0.050
Median	34.82 ± 18.58	34.85 ± 19.89	0.985
Superficial peroneal	10.38 (5.50–14.02)	10.36 (5.15–14.12)	0.795
Sensory CV (m/s)			
Ulnar	51.45 ± 6.24	51.38 ± 5.95	0.896
Median	51.00 ± 7.58	50.84 ± 7.59	0.780
Superficial peroneal	44.35 ± 5.44	44.08 ± 5.30	0.545
F-wave minimum latency (ms)	44.10 ± 4.81	45.21 ± 4.77	0.004
MNAmp	11.08 ± 2.78	10.98 ± 3.04	0.685
MNCV	47.87 ± 4.36	47.06 ± 4.42	0.022
SNAmp	27.20 (18.34–36.96)	25.51 (19.29–34.91)	0.648
SNCV	49.30 ± 5.00	49.74 ± 8.42	0.466

SUA, serum uric acid; CV, conduction velocity; MNAmp, mean motor nerve amplitude; MNCV, mean motor nerve conduction velocity; SNAmp, mean sensory nerve amplitude; SNCV, mean sensory nerve conduction velocity.

**Table 5 brainsci-12-01704-t005:** Correlation analysis of serum uric acid (SUA) with nerve conduction studies parameters.

	Total Patients (*n* = 647)		Male Subgroup (*n* = 404)		Female Subgroup (*n* = 243)		Younger Subgroup (*n* = 449)		Older Subgroup (*n* = 198)	
	r	*p*	r	*p*	r	*p*	r	*p*	r	*p*
Motor amplitude (mV)										
Ulnar	−0.023	0.553	−0.036	0.470	0.045	0.484	−0.043	0.361	0.026	0.714
Median	−0.045	0.252	−0.079	0.112	−0.040	0.537	−0.033	0.483	−0.041	0.565
Tibial	−0.060	0.128	−0.101	0.042	0.031	0.629	−0.042	0.379	−0.079	0.270
Common peroneal	−0.018	0.652	−0.054	0.287	0.026	0.692	−0.011	0.814	0.004	0.960
Motor CV (m/s)										
Ulnar	−0.097	0.013	−0.062	0.215	−0.022	0.733	−0.128	0.007	−0.037	0.601
Median	−0.070	0.075	−0.096	0.054	0.003	0.965	−0.077	0.102	−0.046	0.525
Tibial	−0.104	0.008	−0.152	0.002	0.010	0.883	−0.107	0.023	−0.087	0.229
Common peroneal	−0.104	0.009	−0.105	0.037	0.044	0.501	−0.141	0.003	−0.019	0.792
Sensory amplitude (uV)										
Ulnar	−0.101	0.011	−0.081	0.107	0.044	0.497	−0.130	0.006	−0.027	0.705
Median	−0.007	0.861	−0.013	0.797	0.042	0.516	−0.030	0.528	0.077	0.286
Superficial peroneal	−0.056	0.182	−0.036	0.512	0.042	0.530	−0.042	0.402	−0.007	0.934
Sensory CV (m/s)										
Ulnar	−0.057	0.153	−0.026	0.610	0.023	0.722	−0.075	0.112	−0.010	0.886
Median	−0.005	0.900	−0.047	0.347	−0.048	0.463	−0.019	0.694	0.042	0.562
Superficial peroneal	−0.065	0.124	−0.122	0.024	0.082	0.219	−0.104	0.037	0.016	0.840
F-wave minimum latency (ms)	0.138	0.000	0.171	0.001	−0.086	0.187	0.121	0.010	0.041	0.577
MNAmp	−0.051	0.203	−0.096	0.057	0.026	0.688	−0.038	0.421	−0.047	0.520
MNCV	−0.113	0.004	−0.126	0.012	0.014	0.826	−0.135	0.004	−0.059	0.423
SNAmp	−0.037	0.384	−0.020	0.714	0.098	0.145	−0.072	0.151	0.066	0.405
SNCV	−0.020	0.641	−0.038	0.482	0.030	0.658	−0.061	0.221	0.033	0.679

CV, conduction velocity; MNAmp, mean motor nerve amplitude; MNCV, mean motor nerve conduction velocity; SNAmp, mean sensory nerve amplitude; SNCV, mean sensory nerve conduction velocity.

**Table 6 brainsci-12-01704-t006:** Multivariate linear regression analysis of the relationship between serum uric acid and nerve conduction parameters.

	Total Patients (*n* = 647)		Male Subgroup (*n* = 404)		Younger Subgroup (*n* = 449)	
	β (95% CI)	*p*	β (95% CI)	*p*	β (95% CI)	*p*
MNCV	−0.006 [(−0.009)–(−0.002)]	0.002	−0.007 [(−0.012)–(−0.003)]	0.001	−0.007 [(−0.012)–(−0.002)]	0.004
F-wave minimum latency (ms)	0.007 (0.003–0.011)	0.000	0.01 (0.005–0.015)	0.000	0.006 (0.001–0.011)	0.028
Ulnar Sensory amplitude (µV)	−0.007 [(−0.023)–(0.008)]	0.353	−0.013 [(−0.03)–(0.004)]	0.136	−0.014 [(−0.035)–(0.006)]	0.179
Motor CV (m/s)						
Ulnar	−0.004 [(−0.010)–(0.001)]	0.103	−0.005 [(−0.012)–(0.002)]	0.143	−0.007 [(−0.014)–(−0.001)]	0.034
Median	−0.005 [(−0.010)–(0.000)]	0.032	−0.006 [(−0.012)–(−0.001)]	0.024	−0.008 [(−0.014)–(−0.001)]	0.016
Tibial	−0.008 [(−0.012)–(−0.003)]	0.001	−0.010 [(−0.015)–(−0.005)]	0.000	−0.007 [(−0.013)–(−0.002)]	0.010
Common peroneal	−0.006 [(−0.010)–(−0.002)]	0.004	−0.009 [(−0.014)–(−0.003)]	0.001	−0.008 [(−0.014)–(−0.002)]	0.005

The model was adjusted for age, sex, duration of diabetes, glycated hemoglobin, body mass index, smoking, hypertension, total cholesterol, triglyceride, high-density lipoprotein cholesterol, low-density lipoprotein cholesterol, and free thyroxine (FT4). MNCV, mean motor nerve conduction velocity; CV, conduction velocity; CI, confidence interval.

## Data Availability

The data that support the findings of this study can be made available from the corresponding author at a reasonable request.

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
