# Peer review of "Association of Elevated Serum Uric Acid with Nerve Conduction Function and Peripheral Neuropathy Stratified by Gender and Age in Type 2 Diabetes Patients"

_brainsci, 2022, doi:10.3390/brainsci12121704_

Round 1

Reviewer 1 Report

The aim of the article was to determine the potential associations of uric acid with peripheral nerve conduction in Chinese patients with T2DM. The authors have performed detailed clinical, neurological and neurophysiological testing in 647 patients with T2DM and determined that uric acid levels were higher in participants with DPN. 

The authors should be commended on performing a study in a large number of participants and including detailed neurological and neurophysiological testing, although sural nerve investigation was not included.

There are a number of limitations of the study:

1. The rationale for the study is unclear, the authors state that there are other studies and a meta-analysis which state the uric acid is raised in those with DPN. What is the novelty in this study which adds to the evidence base?

2. The diagnostic criteria for DPN is unclear, they author state that they used the Toronto criteria, but provide no further details. Also, the lack of the sural nerve neurophysiology is unusual.

3. The statistical analysis needs considering, the authors perform a number of statistical analysis and provide a considerable amount of data. However, some is ill conceived, for example uric acid will not be used as a diagnostic test in DPN, therefore ROC is inappropriate. 

4. The interpretation of the results is too strong. At best, at the moment a raised uric acid in DPN is very mild and of questionable clinical significance. The correlation between DPN and uric acid is very weak, and confounding factors are likely to be involved. The authors provide minimal discussion suggesting the potential link between DPN and uric acid.

5. A major concern is the quality of the English throughout. The article needs significant editing before re-submission.

Reviewer 2 Report

The authors investigated the effect of serum uric acid (SUA) level on diabetic peripheral neuropathy (DPN) in patients with type 2 diabetes. The results suggested a correlation SUA level with nerve conduction velocity in same group of patients with type 2 diabetes, thus contributing to the development of DPN.

This is an interesting study suggesting the importance of SUA in the development of diabetic neuropathy. The results of this study provide important insights into current knowledge on the pathophysiology of this disease and contribute to the development of targeted therapy directed against SUA. Considering the prevalence of diabetic neuropathy, this manuscript will attract broad range of readers. I do not have any critical comments.

1. Fig. 1 perfectly illustrates the patient selection process. However, I propose to increase the resolution of Fig.1. Currently resolution is poor quality.

2. The correct performance of the statistical analysis guarantees obtaining the correct result. I would like to ask if the normal distribution was checked before choosing the Student t-test? This should be included in the text of MS.

3. Line 143: the period after the bracket is unnecessary. Moreover, the space before the period at the end of the sentence is unnecessary.

4. Spaces are missing in some parts of the text. Lines: 87, 101, 107, 127, 131, 142, 146, 157, 180, 182, 197, 204, 212, 213, 230, 246: no spaces between some words or parentheses.

5. No spaces between some values in Table 1 and Table 2. It is necessary to standardize the spelling.

6. I propose to increase the resolution of Fig. 2. Currently it is poor quality.

7. I propose to make a Figure comparing the parameters contained in Table 4. The Figure is easier to interpret.

8. Although the number of examined patients is mentioned in the main text, it should also be incorporated in Table 3, 5 and 6. Especially the number of patients in the subgroups.

9. The research is valuable, it relates to DPN in patients with type 2 diabetes. Are there studies that describe the serum uric acid (SUA) level in patients with type 1 diabetes. In the Discussion section, SUA level and its importance in the progression of DPN in patients with type 1 diabetes should be mentioned.

10. Have you performed skin biopsy on patients with DPN? In DPN, skin biopsy can detect abnormalities of the target nerves and vessels.

Reviewer 3 Report

Uric acid is a known risk factor for macrovascular disease. In this paper the authors show from cross-sectional, observational data, an association between serum uric acid and diabetic peripheral neuropathy. 

Any simple measure that alerts clinicians to the possibility of underlying neuropathy in patients with diabetes would be useful. Uric acid is easy to measure, can be performed cheaply and reliably so this paper is worthy of record.

Author Response

Thank you very much for your careful review.

Reviewer 4 Report

Maybe a deeper discussion, with biochemycal explanation and hypothetical mechanisms, might improve the soundness of the paper.

If the hypothesis of sexual hormones might explain the failed association between SUA level and DPN, this should be more extensively described. A further analysis between fertile and menopausal females might help to sustain the findings.

line 125: patients , not patitens

Round 2

Reviewer 5 Report

As for Point 9, "our study lends support ... to prevent the occurrence and development of DPN." I recommend to change the statement to:

"The study lends support to the viewpoint that special attention to be paid to the male and younger patients with elevated SUA level for their susceptibility to DPN."

I have no further comments.

Author Response

Thank you very much for your careful review. We have corrected it.